# Effect of Cooking and *in vitro* Digestion on the Polyphenols and Antioxidant Properties of *Asparagus officinalis* L. cultivars

**DOI:** 10.3390/foods14132367

**Published:** 2025-07-03

**Authors:** Angela Di Matteo, Antonio Paolillo, Lidia Ciriaco, Juliane Lima da Silva, Stefania De Pascale, Luana Izzo

**Affiliations:** 1Foodlab, Department of Pharmacy, University of Naples Federico II, Via Domenico Montesano 49, 80131 Naples, Italy; angeladimatteo95@gmail.com (A.D.M.); antonio.paolillo@unina.it (A.P.); lidia.ciriaco@unina.it (L.C.); 2School of Chemistry and Food, Federal University of Rio Grande, Av. Itália, Km 8, Rio Grande 96203-900, Brazil; julianelima@furg.br; 3Department of Agricultural Sciences, University of Naples Federico II, Via Università 100, 80055 Portici, Italy; depascal@unina.it

**Keywords:** asparagus, bioactive compounds, UHPLC-Q-Orbitrap HRMS, cooking, bioaccessibility

## Abstract

Asparagus (*Asparagus officinalis* L.) is widely recognized for its nutritional and functional properties, attributed to its rich content of polyphenols and antioxidant compounds. However, the content of compounds that remains bioaccessible following typical domestic preparation and digestion remains unclear. This study assessed the polyphenolic profile and antioxidant capacity of the edible portion of two *A. officinalis* cultivars (*Placoseps* and *Darlise*), harvested in different seasons, in edible form, cooked (using boiling on an induction cooktop), and cooked-digested extracts. Rutin emerged as the most abundant in all analyzed samples; its concentration in the edible part reached 1770.72 in *Placoseps* and 995.20 mg/kg in *Darlise*. Cooking increased rutin content in April-harvested asparagus to 1966.00 in *Placoseps* and 2042.44 mg/kg in *Darlise*, reflecting an increase of more than 2.5-fold compared to the respective values observed at the earlier harvest. Despite the substantial reduction in bioactive compounds observed during in vitro gastrointestinal digestion, a total of 146.95 to 454.58 mg/kg of bioaccessible compounds remaining available for potential intestinal absorption after digestion across both cultivars and harvest periods. These results provide a greater understanding of the behavior of polyphenol-rich vegetables and underscore the importance of simulating gastrointestinal processes when assessing the health-promoting potential of bioactive compounds.

## 1. Introduction

*Asparagus officinalis* L. is a perennial plant, native to the eastern Mediterranean and Asia, cultivated for more than two thousand years in more than 60 countries around the world, including Italy, Netherlands, Germany, United States, Canada, among others [1]. Asparagus can be classified by its color into purple-green, green, white, purple-blue, and pink [2]. After the harvesting process, the white asparagus exposed to sunlight is converted into green, which is the most commonly consumed, but white varieties are ordinarily eaten and relished in some countries, like Belgium, Netherlands, and Peru [3]. Among the numerous cultivated varieties of *Asparagus officinalis* L., *Placoseps* and *Darlise* stand out for their distinct agronomic and morphological characteristics. *Placoseps* is widely recognized for its tight apical closure and uniform spear formation, features that enhance its commercial appeal in the fresh vegetable market. *Darlise*, on the other hand, is valued for its robust growth habit, and high productivity, making it particularly suitable for early-season harvests [4].

Global production of “vegetables, other” reached approximately 1 billion tonnes (1,006,227,000 t), emphasizing the significance of these crops in the agricultural and food sector. Europe contributed around 58.5 million tonnes, accounting for 5.8% of global production. Italy, with approximately 5 million tonnes, represented 8.6% of Europe’s total output, ranking fourth after the Russian, Spain, and Ukraine [5]. Data acquired through the Fourth Italian National Food Consumption Survey (IV SCAI) reported the Italian consumption of all other types of fresh vegetables. Asparagus included in “other vegetables, fresh” food group revealed a mean of intake of 0.4 g/day (on total population; n = 1969), whereas considering the consumer the daily intake corresponds to 1.3% (23.7 g/day). The consumption of vegetables was strongly varied between elderly and adults compared to the younger subjects [6].

It is appreciated and widely consumed not only for its delicate taste and texture but also for its nutritional values and functional health benefits [7]. Asparagus is notable for its high levels of protein (approximately 2 g/100 g), low fat content (around 0.1 g/100 g), and significant amounts of vitamins and minerals—such as vitamin K (41.6 µg), vitamin C (9.2 mg), niacin (1.08 mg), potassium (278 mg), phosphorus (54 mg), and calcium (21 mg) per 100 g—which can be up to five times greater than those found in many common vegetables [8].

Additionally, asparagus contains nine essential amino acids such as lysine, leucine, isoleucine, and threonine, providing a complete amino acid profile. This confers a higher biological value compared to that of the majority of other vegetables [9].

Since ancient times, asparagus has been employed not only as a food, but also for its positive impacts on human health, reported also in traditional Chinese medicine as a medicinal herb [10,11]. Actually, asparagus is considered a significant source of bioactive compounds, as polysaccharides, saponins, flavonoids, dietary fibers and oligosaccharides [12], with different physiological effects, including scavenger and antioxidant power [13], as well as diuretic [7], antiepileptic effects [14], hypolipidemic [15,16], decongestant [17], immunomodulatory [18], antifungal [19], anticancer, and antiproliferative [20,21,22]. Polyphenols, secondary metabolites naturally present in foods, represent the most significant class of bioactive compounds of asparagus [2]. In detail, phenolic compounds reported in asparagus, include quercetin, cinnamic acid, rutin, ferulic acid, kaempferol, caffeic acid, gallic acid, and others [23], with rutin being the predominant compound [24,25]. Flavonoids include quercetin, isorhamnetin, and kaempferol, although present in smaller quantities [26]. Finally, the main saponins, isolated from different commercial varieties include protodioscin, yamogenin, asparanin A, and sarsasapogenin [27]. Polyphenols have gained increasing attention for their ability to modulate oxidative stress, inflammation, and critical cellular signaling pathways involved in the development of chronic diseases. Their potent antioxidant activity has been linked to a decreased risk of cardiovascular diseases, neurodegenerative disorders, and certain cancers [28].

Although polyphenols are commonly considered to be sensitive to unfavorable temperatures and light [29], there are scientific evidence showing that in some cases, for certain vegetables and under particular cooking conditions, certain compounds can even double the polyphenol content [30]. Murador et al. [31] reported that cooking process increases polyphenols through the breakdown of the cellular matrix, whereas boiling can cause their loss through leaching and oxidation, especially with prolonged times. Previous work on asparagus also assessed that, on average across cultivars, cooking resulted in an increase in the levels of total phenols [32]. In the study proposed by Andlauer et al. [33], zucchini, carrots, and green beans retained higher amounts of rutin and chlorogenic acid when cooked with a small amount of water. In carrots, chlorogenic acid decreased from 82.4 to 29.6 mg/kg with lots of water but to only 60.3 mg/kg with little water [34].

The beneficial effects on human health due to the intake of phenolic compounds are only exerted when they are bioaccessible and bioavailable within the gastrointestinal and circulatory systems [35]. During gastrointestinal digestion, bioactive compounds may undergo significant molecular modifications because of various physiological factors. These include the highly acidic environment of the stomach, which can lead to structural degradation or transformation of sensitive molecules, as well as the sequential action of digestive enzymes. Such conditions can alter the stability, solubility, and overall bioaccessibility of compounds, ultimately influencing their biological capacity and potential absorption [36]. Despite increasing interest in the polyphenolic composition of asparagus, limited data are available regarding the bioaccessibility of individual compounds following gastrointestinal digestion. The quantification of phenolic compounds before and after in vitro digestion helps to understand the real impact of asparagus polyphenols on human health.

In addition, environmental conditions and harvest season significantly influence the biosynthesis and accumulation of polyphenols in asparagus spears. In Italy, the February harvest corresponds to the early harvest period, particularly in southern regions, while April represents a late harvest stage for the *Placosep* and *Darlise* asparagus cultivars. The inclusion of two distinct harvest periods enables the assessment of seasonal influences on the polyphenol composition of asparagus, offering novel insights into the impact of early- and late-harvested crops [4,37].

This integrated approach—considering cultivar, cooking, and digestion—offers a comprehensive evaluation of asparagus as a functional food, with implications for nutrition science, food processing, and the agri-food industry, particularly in supporting cultivar selection based on nutritional and functional properties. Within this framework, the current study aims to (i) evaluate the polyphenolic profile and antioxidant activity of two different cultivar of A. officinalis (*Placoseps* and *Darlise*), (ii) investigate the impacts of cooking process on the content of bioactive and finally, and (iii) subject the cooked asparagus to an in vitro gastrointestinal digestion to investigate the total polyphenols remaining available for potential intestinal absorption after digestion.

## 2. Materials and Methods

### 2.1. Chemicals and Reagents

Water (LC-MS grade), methanol, hydrochloric acid, and formic acid were purchased from Merck (Darmstadt, Germany). Polyphenolic standards (≥98% purity) included protocatechuic acid, myricetin, chlorogenic acid, epicatechin, caffeic acid, luteolin, *p*-coumaric acid, vitexin, quinic acid, apigenin-7-O-glucoside, catechin, ferulic acid, naringin, quercetin-3-galactoside, diosmin, quercetin, kaempferol-3-glucoside, isorhamnetin-3-rutinoside, daidzein, naringenin, rutin, kaempferol, naringenin, genistein, gallic acid, and apigenin were supplied by Sigma-Aldrich (Milan, Italy).

Potassium persulfate, ferric chloride, sodium acetate, and reference standards of 2,2′-azino-bis (3-ethylbenzothiazoline-6-sulfonic acid (ABTS), and 6hydroxy-2,5,7,8-tetramethylchroman-2-carboxylic acid (Trolox^®^) were obtained from Sigma-Aldrich (Milan, Italy).

Enzymes used for in vitro digestion simulation included α-amylase from human saliva exhibited an activity of at least 1000–3000 U/mg, pepsin, with an enzymatic activity of at least 2500 units per milligram from porcine gastric mucosa, pancreatin from porcine pancreas (8 × USP), and Pronase E (≥3.5 U/mg) from *Streptomyces griseus*, were acquired from Sigma-Aldrich (Milan, Italy). Other digestion components included calcium chloride dihydrate (CaCl_2_·2H_2_O), potassium chloride, potassium thiocyanate, monosodium phosphate, sodium chloride, sodium sulfate, sodium hydroxide, and sodium bicarbonate were sourced from Sigma-Aldrich (Milan, Italy).

Syringe filters (Phenex-NY 0.2 µm, 15 mm), amber vials, and polypropylene conical centrifuge tubes were obtained from Phenomenex (Castel Maggiore, Italy) and Carlo Erba (Milan, Italy), respectively.

### 2.2. Sample Collection and Preparation

Asparagus (*Asparagus officinalis* L.) from two cultivars, *Placoseps* and *Darlise*, were harvested in February and April 2024 from F.lli Laezza (Societa’ Semplice Agricola) located in Acerra (Campania, Italy). The analysis was conducted on three different sampling conditions: edible, cooked, and cooked-digested portions. The edible portions refer to the raw edible parts of asparagus spears, including the apical and green sections, obtained after removal of the basal segments. The cooked portions consisted of the edible parts of asparagus spears subjected to cooking using an induction hob. The cooked-digested portions, consisting of cooked samples, were further subjected to a standardized in vitro gastrointestinal digestion procedure. Upon arrival at the laboratory, the raw samples were thoroughly washed with distilled water to remove soil and impurities, dried with inert cloths, and trimmed to a standardized length of 16 cm. The samples were freeze-dried (−20 °C) and finely ground before analysis. The dry matter content of both raw and cooked samples was about 9% (*w*/*w*). The freeze-dried material was finely ground using a laboratory mill and stored in 50 mL polypropylene tubes until analysis.

### 2.3. Polyphenolic Compound Extraction

Polyphenols were extracted using a modified method based on Castaldo et al. [38]. 1 g of lyophilized asparagus was suspended in 20 mL of H_2_O and incubated at 40 °C for 30 min in a shaking water bath at 120 rpm (1086, GFL, Rome, Italy). The mixture was then sonicated for 30 min at 4 °C. After centrifugation at 5000× *g* for 5 min, the supernatant was collected. The remaining pellet was re-extracted under same conditions, and the supernatants were pooled. Aliquots were filtered through nylon syringe filters (0.2 µm), acidified with 0.1% formic acid in methanol, and stored for UHPLC-Q-Orbitrap HRMS analysis (Dionex, Thermo Fisher Scientific, Waltham, MA, USA).

### 2.4. Ultra-High Liquid Chromatography Coupled to Mass Spectrometry Analysis

Polyphenolic profiling was carried out using an UltiMate 3000 UHPLC system (Dionex, Thermo Fisher Scientific, Waltham, MA, USA) equipped with a Kinetex Biphenyl column (100 × 2.1 mm, 2.6 µm; Phenomenex, Torrance, CA, USA) maintained at 30 °C. The chromatographic separation employed a binary solvent system consisting of 0.1% formic acid in water (solvent A) and 0.1% formic acid in methanol (solvent B). The gradient elution was as follows: 100% A for 0.5 min, linear decrease to 30% A over 1 min, held for 6.5 min, then reduced to 15% A for 3 min, and finally returned to initial conditions within 2 min. The mobile phase flow rate was set at 0.5 mL/min and the injection volume was 5 µL.

Mass spectrometric detection was performed using a Q-Exactive Orbitrap (Thermo Fisher Scientific) equipped with an electrospray ionization source operating in negative mode. Data acquisition included two scan events: all ion fragmentation (AIF) and full ion MS. AIF parameters were configured as follows: resolution of 17,500 full widths at half-maximum (FWHM), automatic gain control (AGC) target 1 × 10^5^, scan time 0.10 s, maximum injection time to 200 ms, isolation window to 5 *m*/*z*, scan range 80–1200 *m*/*z*, and retention time to 30 s. The collision energies were included in the range between 15 and 45 eV. For full MS, the resolution was set at 70,000 FWHM, with a scan range 80–1200 *m*/*z*, injection time set to 200 ms, AGC target 1 × 10^6^, and scan rate at 2 scan/s. Ion source settings included a capillary temperature 275 °C, S-lens RF level 50, spray voltage 2.8 kV, sheath gas pressure (N_2_ > 95%) 35, auxiliary gas (N_2_ > 95%) 10 and auxiliary gas heather temperature 350 °C. Mass accuracy was ensured by applying tolerance of 5 ppm.

The raw data were processed and analyzed using Xcalibur software (version 3.1.66.19; Xcalibur, Thermo Fisher Scientific, Waltham, MA, USA).

### 2.5. Cooking Procedure

Edible portion of asparagus from both cultivars was subjected to mild cooking to simulate common domestic preparation. Specifically, 100 g fresh samples were boiled with 250 mL of water. The induction hob (model Y63IV443, Electrolux, Stockholm, Sweden) was set to power level 10, corresponding to approximately 3.5 kW and a temperature of 100 °C, ensuring rapid and consistent boiling. The samples were cooked for 10 min. During cooking, the water was completely absorbed by the asparagus, ensuring no residual liquid remained. Cooked samples were immediately frozen –20 °C, lyophilized, and subjected to further analysis.

### 2.6. Simulated In Vitro Digestion

In vitro gastrointestinal digestion was simulated following the standardized INFOGEST protocol [39], which mimics human oral, gastric, and intestinal digestion phases. For the oral phase, 5 mL of cooked asparagus extract were mixed with 3.5 mL of simulated salivary fluid, 0.5 mL of α-amylase solution, 975 µL of deionized water, and 25 µL of 0.3 M calcium chloride. The mixture was incubated at 37 °C for 2 min in a shaking water bath (100× *g*). Then, for the gastric phase, 7.5 mL of simulated gastric fluid, 1.6 mL of pepsin solution (2000 U/mL), 695 µL of water, and 5 µL of 0.3 M calcium chloride were added. The pH was adjusted to 3 using 1 M HCl, and the sample was incubated at 37 °C for 2 h under constant agitation.

Then, for the intestinal phase, the gastric mixture was supplemented with 11 mL of simulated intestinal fluid, 5 mL of pancreatin solution (100 U/mL trypsin activity), 2.5 mL of bile salt solution (65 mg/mL), 40 µL of 0.3 M calcium chloride deionized water was added to reach a total volume of 20 mL. The pH was raised to 8 with 1 M NaOH, and the digestion proceeded for a further 2 h at 37 °C with agitation (100× *g*) in an orbital shaker bath. After digestion, the samples were centrifuged at 5000× *g* for 10 min, and the supernatant was freeze-dried at −20 °C for further analysis.

### 2.7. Antioxidant Activity Assays

Antioxidant capacity was assessed in the edible portion, cooked samples, and digested samples of asparagus from both *Placoseps* and *Darlise* cultivars, harvested in February and April 2024. The assays used were DPPH, ABTS, and FRAP. Results were expressed as mmol Trolox equivalents (TE) per kilogram of dry weight extracts.

#### 2.7.1. ABTS Assay

The ABTS radical-scavenging capacity was evaluated following the method described by Izzo et al. [40]. Briefly, 2.5 mL of a 7 mM ABTS solution was mixed with 44 µL of a 2.5 mM potassium persulfate solution. The resulting solution was incubated at room temperature for 16 h, protected from light. The working solution was diluted with ethanol to reach an absorbance of 0.70 ± 0.05 at 734 nm. A 100 µL aliquot of diluted extracts was added to 1 mL of ABTS solution, and absorbance was recorded at 734 nm, after 3 min of incubation.

#### 2.7.2. DPPH Assay

The DPPH radical scavenging activity was assessed as described by Brand-Williams et al. [41]. A total of 1 mg of DPPH was solubilized in methanol to obtain a solution with an absorbance of 0.90 (±0.02) at 517 nm. Subsequently, 200 µL of each appropriately diluted sample was added to 1 mL of the DPPH working solution, and the mixture was incubated. The absorbance was recorded after 10 min in dark conditions at 517 nm.

#### 2.7.3. FRAP Assay

Ferric reducing antioxidant power was assessed according to Rajurkar and Hande [42]. The FRAP working solutions were prepared by combining 1.25 mL of FeCl_3_ (20 mM), 1.25 mL of TPTZ (10 mM in 40 mM HCl), and 12.5 mL of acetate buffer (0.3 M, pH 3.6). A volume of 2.85 mL of the reagent was then mixed with 150 µL of diluted sample, and the resulting absorbance at 593 nm was measured after a 4-min incubation.

### 2.8. Total Phenolic Content

Total phenolic content was quantified using the Folin–Ciocalteu colorimetric method according to Tenore et al. [43]. A 125 µL aliquot of diluted extracts was added to 500 µL of water and 125 µL of 2 N Folin–Ciocalteu reagent. After 6 min, 1.25 mL of 7.5% sodium carbonate and 1 mL of deionized water were added. Following incubation in the dark at room temperature for 90 min, absorbance was measured at 760 nm. Total phenolic content was expressed as milligrams of gallic acid equivalents (GAE) per gram of dry extract.

### 2.9. Statistical Analysis

Data analysis was carried out using two-way analysis of variance (ANOVA) in SPSS version 13.0 https://spss.software.informer.com/13.0/ (accessed on 30 June 2025). Significant differences among means were identified using the Tukey–Kramer multiple comparison test; with *p*-values ≤ 0.05 considered statistically significant. All measurements were performed in triplicate, and the results were presented as mean ± standard deviation (SD).

## 3. Results

### 3.1. Chemical Characterization of Polyphenols in Aqueous Extract of Asparagus by UHPLC Q-Exactive

Aqueous extracts from the edible portion, cooked portion, and cooked-digested portion of *Asparagus officinalis* L., from both *Placoseps* and *Darlise* cultivars and two harvest periods (February and April), were analyzed using UHPLC Q-Orbitrap HRMS to identify up to 27 phenolic compounds, comprising both flavonoids and phenolic acids. The analytical method allowed the optimal chromatographic separation within a total run time of 13 min. Mass spectrometric parameters, including ion assignment, retention time, theoretical and measured mass (*m*/*z*), product ions, mass sensitivity and accuracy, are reported in Table 1. All assays were conducted in negative electrospray ionization (ESI^−^) mode. All phenolic compounds were identified based on a combination of retention time, MS/MS fragmentation patterns and comparison with authentic standards. Structural identification of isomeric compounds, including catechin and epicatechin (*m*/*z* 289.07199); apigenin and genistein (*m*/*z* 269.04555); kaempferol and luteolin (*m*/*z* 285.04062); kaempferol-3-O-glucoside and luteolin-7-glucoside (*m*/*z* 447.09360), was performed by comparing their retention times and fragmentation patterns with reference data available in the literature.

### 3.2. Chemical Quantification of Polyphenols in the Aqueous Extract of Asparagus Edible Portions

Targeted quantification of polyphenols in the edible portion of *Asparagus officinalis* L. was performed using a calibration curve constructed over twelve concentration levels. Linearity was excellent for 27 analytes, with regression coefficients (R^2^) greater than 0.99. The quantitative results findings are presented in Table 2 and reported as milligrams per kilogram (mg/kg) of extract. The polyphenolic composition showed considerable variation among cultivars and harvest periods. Among the identified compounds, rutin consistently emerged as the most abundant in all analyzed samples. In April, its concentration reached 1770.72 mg/kg in *Placoseps* and 995.20 mg/kg in *Darlise*, representing around 70% of the total polyphenol content in both cultivars. Seasonal differences were evident across all major flavonols. In *Placoseps*, rutin levels increased by over 100-fold from February to April, while in *Darlise*, the increase was approximately six-fold. Similarly, isorhamnetin-3-rutinoside content increased by about five times in *Placoseps* and over six times in *Darlise* across the same period. Isoquercetin content showed a slight increase in *Placoseps* from February to April, whereas the increase was more marked in *Darlise*. Among phenolic acids, quinic acid and ferulic acid were the most represented. Quinic acid was the predominant phenolic acid across all samples, with concentrations ranging from 214.00 to 327.12 mg/kg. Its content slightly decreased in April (approximately −28% in *Placoseps* and –19% in *Darlise*), compared to February in both cultivars. Regarding ferulic acid concentration, results show an increase from February to April 2024, with an approximate rise of 106% in *Placoseps* (from 36.76 ± 0.06 to 75.68 ± 0.11 mg/kg) and 260% in *Darlise* (from 11.72 ± 0.51 to 42.24 ± 1.30 mg/kg).

### 3.3. Chemical Quantification of Polyphenols in the Aqueous Extract of Asparagus Cooked Portions

The polyphenolic profile of asparagus samples was further investigated following cooking to investigate the impact of thermal treatment on the chemical composition and relative abundance of individual phenolic compounds. In the cooked fractions, flavonols remained the predominant class of polyphenols, amounting to rep 79% of total polyphenolic content in both cultivars at the later harvest stage; this pattern was largely attributable to the high concentrations of rutin. The cooking of the asparagus matures showed rutin content at 1966.00 mg/kg in *Placoseps* and 2042.44 mg/kg in *Darlise*, reflecting an increase of more than 2.5-fold compared to the respective values observed at the earlier harvest. Isorhamnetin-3-rutinoside also showed a marked increase over the harvest period, particularly in the *Darlise* cultivar, where its concentration increased by over 200%. Isoquercetin levels increased from the edible to the cooked samples, with a more noticeable rise observed in the February harvest for both cultivars. Among phenolic acids, chlorogenic acid exhibited a pronounced increase after thermal treatment, especially in *Darlise*, where its content tripled between February and April. A substantial increase was also observed in *Placoseps*. On the other hand, quinic acid, considered one of the more stable phenolic acids, displayed cultivar-specific trends: its levels slightly decreased in *Placoseps* and in *Darlise* over the same time frame. Ferulic acid concentrations were overall lower in the cooked samples compared to the edible extracts, although a slight increase was observed after the first harvest in *Darlise*. Naringin, the only detected flavanone, showed a modest decrease across the two sampling points in both cultivars. Flavones were consistently found at trace levels, remaining below the limit of quantification (<LOQ), as reported in Table 3.

### 3.4. Chemical Quantification of Polyphenols in the Aqueous Extract of Digested Asparagus Portions

A cooked portion of both asparagus varieties, harvested at two different times, was subjected to in vitro simulated digestion to assess the impact of digestion on polyphenols and their inaccessibility. Results are reported in Table 4 and expressed as mg per kilogram of extract. Overall, digestion significantly reduced both the total polyphenol content and individual polyphenolic classes across both cultivars and harvest periods. A comparison between the cooked samples and cooked-digested samples indicates that: phenolic acids concentration from the April harvest decreased by approximately 4-fold in both cultivars. Among this class, quinic acid remained the most abundant compound. A comparable trend was observed for total flavonols, which declined from 6-fold to 13-fold in both cultivars and harvest periods. Rutin consistently represents the predominant compound with around 30% and 50% total flavonols, respectively, in the February and April harvests. Polyphenols’ bioaccessibility after gastrointestinal stage was between 12.5 and 18.9% across both cultivars and harvesting stages.

### 3.5. Analysis of TPC in the Aqueous Extract of Asparagus officinalis

The total phenolic content (TPC) was quantified using the Folin–Ciocalteu colorimetric method, a widely adopted technique to evaluate the concentration of phenolic compounds in biological matrices. TPC values are reported as mg of gallic acid equivalents per gram of sample. Results are reported as mean values ± standard deviation (SD), based on analytical replicates for each sample. Analysis of the TPC in the edible portions of asparagus from two cultivars, *Placoseps* and *Darlise*, harvested in February and April, reveals a significant increase in phenolic content from the earlier to the later harvest for both cultivars. This indicates a seasonal effect on phenolic accumulation, with samples harvested in April showing markedly higher TPC. The total polyphenol content in the cooked portions exhibited an increase compared to the edible parts, consistently observed across both cultivars and harvest periods. Conversely, following in vitro digestion, the total polyphenol content decreased, in agreement with the trends observed for individual polyphenolic compounds. TPC assay demonstrated a decrease in total polyphenols content of around 40% compared to the results of the edible (7.01 and 5.89 mg GAE/g) versus cooked digestion (4.24 and 3.61 mg GAE/g) harvest in April for both cultivars, revealing a substantial loss in polyphenol bioaccessibility. The trend of higher TPC in April relative to February is preserved even after cooking and digestion (Table 5).

### 3.6. Antioxidant Assays of the Aqueous Extract of Asparagus officinalis

The antioxidant capacity of asparagus extracts was evaluated using three different assays: ABTS, DPPH, and FRAP. Results are reported as millimoles of Trolox equivalent per kilogram of dry extract, reported as mean values ± standard deviation (Table 6).

In the edible portion, antioxidant activity increased substantially from February to April, particularly in the *Placoseps* cultivar. Specifically, DPPH values increased from 6.74 to 18.31 mmol Trolox/kg, while FRAP values rose from 13.26 to 21.11 mmol Trolox/kg. The *Darlise* cultivar showed more moderate changes, with a 44% increase in DPPH activity and a relative stable FRAP response across the two harvest periods.

Following thermal treatment, antioxidant levels generally increased compared to the edible extracts. In *Placoseps*, DPPH values increased from 10.23 to 23.71 mmol Trolox/kg between February and April (+132%), and ABTS values rose from 53.65 to 67.69 mmol Trolox/kg (+27%). In *Darlise*, the DPPH increased from 12.13 to 21.69 mmol Trolox/kg (+59%), while ABTS values increased moderately from 51.72 to 59.88 mmol Trolox/kg (+16%). FRAP values followed a similar trend in both cultivars, with a 62% increase in *Placoseps* and 30% in *Darlise* across the two harvest times. Following in vitro digestion, all assays revealed a decrease in the antioxidant activity compared to both the edible and cooked samples. Notably, samples from the April harvest exhibited higher antioxidant activity than those from February, a trend consistent across both cultivars. These results align with the observed reductions in total and individual polyphenol contents post-digestion, further supporting the close relationship between polyphenol composition and antioxidant capacity. Moreover, antioxidant activity measured by FRAP, DPPH, and ABTS assays were correlated with TPC values (Appendix A).

## 4. Discussion

The comprehensive profiling of polyphenolic compounds in *Asparagus officinalis*, evaluated across multiple harvest periods and cultivars with a focus on the edible portion after cooking and simulated gastrointestinal digestion, provides critical insights into the compositional changes, processing-induced modifications, and the bioaccessibility of these phytochemicals. This integrative approach allows for a deeper understanding of how both technological treatments (such as thermal processing) and physiological factors (including digestive processes) influence the nutritional and functional properties of asparagus.

Flavonols, such as rutin, were consistently identified as the most abundant polyphenolic subclass, in line with previous findings that report rutin as the major flavonoid in asparagus spears [44,45]. A clear seasonal and genotypic effect was observed: April-harvested extracts exhibited higher levels of total polyphenols compared to those collected in February, a difference likely reflecting a more advanced stage of phenolic metabolism associated with increased environmental exposure to light and temperature and ripening [46,47]. Additionally, cultivar-specific traits appeared to influence polyphenol accumulation. *Placoseps* showed higher overall levels of phenolic compounds, plausibly due to enhanced biosynthesis of secondary metabolites associated with its pigmentation [48]. In contrast, *Darlise*, a cultivar characterized by earlier spear development and milder morphological traits, generally exhibited lower phenolic concentrations [4].

Thermal processing generally resulted in an improvement in total phenolic content (TPC). This increase was particularly evident for specific individual polyphenols, notably chlorogenic acid and rutin, especially in samples harvested in April. The increased extractability of polyphenols is thought from heat-induced disruption of cell wall integrity and subsequent hydrolysis of glycosidic and ester linkages [49,50,51]. Nonetheless, thermal degradation was evident for some compounds such as quinic acid, which decreased in cooked samples, for both cultivars. Similarly, ferulic acid generally declined after cooking, except for a marked increase observed in *Darlise* harvested in February, suggesting a partial release from bound forms under specific processing conditions [32].

Although cooking appeared to enhance the release of certain polyphenols, this does not necessarily reflect their actual bioaccessibility. To better assess this aspect, the present study evaluated polyphenol content after in vitro digestion of cooked edible portion. A marked reduction in total polyphenolic concentration was observed across all samples following digestion, supporting the hypothesis that matrix interactions, enzymatic hydrolysis, pH variations, and oxidative conditions during the digestive process significantly affect the integrity and recoverability of polyphenols. Moreover, several studies evaluated the interactions of polyphenols with food constituents such as carbohydrates, lipids, proteins, and fiber, highlighting their significant influence on polyphenol content [52,53].

For example, in the February-harvested *Placoseps* sample, total phenolics decreased from 1174.29 mg/kg after cooking to 146.95 mg/kg post-digestion, corresponding to an 87.5% reduction. Comparable trends were observed across both cultivars and harvest periods, suggesting that a considerable portion of the polyphenols released by cooking is not preserved in a bioaccessible form following digestion. However, following digestion, a slightly greater quantity of phenolic compounds was observed in *Darlise* compared to *Placoseps*.

To our knowledge, no prior studies have systematically assessed the bioaccessibility of polyphenols in *Asparagus officinalis* following simulated gastrointestinal digestion. However, our findings are in agreement with previous research conducted on other vegetables. For instance, Tagliazucchi et al. [54], documented significant reductions in phenolic content during the digestion of tomato matrices, particularly during the intestinal phase, where a shift to neutral-to-alkaline pH conditions and enzymatic degradation contributed to phenolic instability. Likewise, Vallejo et al. [55] also documented extensive degradation and structural modifications of phenolic acids and flavonoids in broccoli during digestion, limiting their potential bioavailability. Similar outcomes were observed in quinoa and djulis sprouts, where Zhang et al. [56] reported marked reductions in total phenolic content and antioxidant activity during in vitro gastrointestinal digestion, despite an initial release of bound compounds.

Antioxidant capacity assays supported the trends observed in polyphenol quantification [57]. Asparagus samples harvested in April consistently exhibited higher antioxidant activity in comparison to those from February, mirroring the elevated flavonol content. These findings confirm that both the quantity and quality of antioxidant compounds are influenced by processing and harvest time [58,59]. The impact of the cultivar was also evident in the results. *Placoseps* samples showed a more pronounced increase in total polyphenols and antioxidant capacity across the season than Darlise, suggesting a greater responsiveness to environmental and developmental factors. This varietal difference aligns with the previous literature reporting genotype-specific accumulation patterns of flavonoids and phenolic acids in asparagus and other vegetable crops [4].

## 5. Conclusions

The analysis revealed that asparagus is a rich source of chlorogenic acid, rutin, ferulic acid, quinic acid, and a range of other flavonoid compounds, highlighting its potential as a functional food with antioxidant properties. Thermal processing may enhance the release of specific compounds such as rutin and chlorogenic acid. Rutin content reached 1966.00 in *Placoseps* and 2042.44 mg/kg dw in *Darlise* in the cooked samples harvested in April. Our results highlighted that mild and limited time cooking helps preserve thermal degradation of phenolic compounds.

Through targeted profiling and quantification, it becomes evident that while thermal processing may enhance the release of specific compounds, this does not necessarily lead to greater bioaccessibility following digestion. Despite the substantial reduction in bioactive compounds observed during in vitro gastrointestinal digestion, a total of 146.95 to 454.58 mg/kg dw of bioaccessible compounds was retained across both cultivars and harvest periods. The observed seasonal differences, with higher bioactive compound levels in the later harvest (April), suggest that the timing of harvest may significantly influence the functional quality of asparagus, providing a basis for recommending later-season consumption for improved polyphenol intake.

From an applied perspective, these insights could inform breeding strategies for more bioaccessible cultivars, guide the development of novel food processing techniques aimed at enhancing polyphenol stability. Future studies should further investigate the mechanisms underlying polyphenol degradation during digestion but also evaluate bioavailability in order to measure the active fraction that reach unaltered the systemic circulation. Moreover, formulation approaches (e.g., use of encapsulants) that may improve the delivery of these health-promoting compounds should be explored.

## Figures and Tables

**Table 1 foods-14-02367-t001:** LC/MS parameters of the investigated compounds (*n* = 27).

Compound	RT (min)	Adduct ion	Chemical Formula	Theoretical Mass (*m*/*z*)	Measured Mass (*m*/*z*)	Accuracy (Δ ppm)	LOD (mg/kg)	LOQ (mg/kg)
Quinic acid	0.64	[M-H]^−^	C_7_H_12_O_6_	191.05528	191.05534	0.314	0.019	0.039
Protocatechuic acid	3.05	[M-H]^−^	C_7_H_6_O_4_	153.01811	153.01819	0.523	0.019	0.039
Quercetin	3.67	[M-H]^−^	C_15_H_10_O_7_	301.03549	301.03549	0.000	0.019	0.057
Chlorogenic acid	3.94	[M-H]^−^	C_16_H_18_O_9_	353.08847	353.08798	−1.388	0.019	0.039
Caffeic acid	4.11	[M-H]^−^	C_9_H_8_O_4_	179.03403	179.03410	0.391	0.013	0.039
Epicatechin	4.12	[M-H]^−^	C_15_H_14_O_6_	289.07205	289.07199	−0.208	0.002	0.004
Catechin	4.16	[M-H]^−^	C_15_H_14_O_6_	289.07196	289.07199	0.104	0.002	0.004
Daidzin	4.32	[M-H]^−^	C_21_H_20_O_9_	415.10388	415.10391	0.072	0.039	0.078
*p*-coumaric acid	4.34	[M-H]^−^	C_9_H_8_O_3_	163.03894	163.03899	0.307	0.013	0.039
Ellagic acid	4.36	[M-H]^−^	C_14_H_6_O_8_	300.99911	300.99905	−0.199	0.002	0.004
Rutin	4.40	[M-H]^−^	C_27_H_30_O_16_	609.14648	609.14697	0.804	0.013	0.039
Isoquercetin	4.43	[M-H]^−^	C_21_H_20_O_12_	463.08862	463.08881	0.410	0.002	0.004
Rosmarinic acid	4.46	[M-H]^−^	C_18_H_16_O_8_	359.07745	359.07773	0.780	0.002	0.004
Ferulic acid	4.49	[M-H]^−^	C_10_H_10_O_4_	193.04964	193.0499	1.347	0.026	0.078
Kaempferol-3-O-glucoside	4.55	[M-H]^−^	C_21_H_20_O_11_	447.09360	447.09372	0.268	0.002	0.004
Luteolin-7-glucoside	4.55	[M-H]^−^	C_21_H_20_O_11_	447.09360	447.09402	0.939	0.004	0.019
Isorhamnetin-3-rutinoside	4.57	[M-H]^−^	C_28_H_32_O_16_	623.16211	623.16260	0.786	0.002	0.004
Myricetin	4.59	[M-H]^−^	C_15_H_10_O_8_	317.03018	317.03027	0.284	0.002	0.004
Naringin	4.69	[M-H]^−^	C_27_H_32_O_14_	579.17242	579.17291	0.846	0.013	0.039
Hesperidin	4.78	[M-H]^-^	C_28_H_34_O_15_	609.18256	609.18317	1.001	0.002	0.004
Diosmin	4.81	[M-H]^−^	C_28_H_32_O_15_	607.16736	607.16803	1.103	0.004	0.009
Luteolin	4.93	[M-H]^−^	C_15_H_10_O_6_	285.04062	285.04062	0.000	0.026	0.078
Daidzein	4.98	[M-H]^−^	C_15_H_10_O_4_	253.05061	253.05057	−0.158	0.004	0.009
Kaempferol	5.11	[M-H]^−^	C_15_H_10_O_6_	285.04062	285.04037	−0.877	0.013	0.039
Naringenin	5.22	[M-H]^−^	C_15_H_12_O_5_	271.06143	271.06125	−0.664	0.002	0.004
Apigenin	5.23	[M-H]^−^	C_15_H_10_O_5_	269.04555	269.04562	0.260	0.013	0.039
Genistein	5.27	[M-H]^−^	C_15_H_10_O_5_	269.04555	269.04556	0.037	0.002	0.039

RT: retention time; LOD: limit of detection; LOQ: limit of quantification.

**Table 2 foods-14-02367-t002:** Polyphenolic content (mg/kg dry weight) in aqueous extracts of edible parts of asparagus spears from two cultivars (*Placoseps* and *Darlise*) harvested in February and April 2024, analyzed by UHPLC Q-Orbitrap HRMS. Values are expressed as mean ± standard deviation.

	Edible Portion (mg/kg) ± SD			
	*Placoseps*		*Darlise*	
**Compound**	**February 2024**	**April 2024**	**February 2024**	**April 2024**
PHENOLIC ACID				
*Cinnamic acid*				
Chlorogenic acid	1.80 ± 0.06	90.24 ± 0.11	10.80 ± 0.11	26.08 ± 0.34
Ferulic acid	36.76 ± 0.06	75.68 ± 0.11	11.72 ± 0.51	42.24 ± 1.30
*p*-Coumaric acid	1.60 ± 0.28	1.34 ± 1.50	2.00 ± 1.13	3.70 ± 0.14
Quinic acid	298.81 ± 5.32	214.00 ± 2.60	327.12 ± 6.01	265.48 ± 1.19
**SUM**	**338.97 ± 5.72**	**381.26 ± 4.32**	**351.64 ± 7.76**	**337.50 ± 2.97**
FLAVONOIDS				
*Flavones*				
Luteolin	*nf*	<LOQ	*nf*	<LOQ
Kaempferol	*nf*	<LOQ	*nf*	<LOQ
**SUM**	**-**	**-**	**-**	**-**
*Flavonols*				
Isoquercetin	29.87 ± 0.04	32.84 ± 1.07	5.44 ± 0.11	29.36 ± 0.00
Rutin	12.13 ± 0.23	1770.72 ± 1.02	164.92 ± 0.06	995.20 ± 5.77
Isorhamnetin-3-rutinoside	29.36 ± 0.00	152.5 ± 0.40	13.88 ± 0.17	81.04 ± 0.34
**SUM**	**71.36 ± 0.27**	**1956.08 ± 2.49**	**184.24 ± 0.34**	**1105.60 ± 6.11**
**TOTAL POLYPHENOLS**	**410.33 ± 5.99**	**2337.34 ± 6.81**	**535.88 ± 8.10**	**1443.10 ± 9.08**

**Table 3 foods-14-02367-t003:** Polyphenolic content (mg/kg dry weight) in aqueous extracts of cooked portions of asparagus spears from two cultivars (*Placoseps* and *Darlise*) harvested in February and April 2024, analyzed by UHPLC Q-Orbitrap HRMS. Values are expressed as mean ± standard deviation.

	Cooked Portion (mg/kg) ± SD			
	*Placoseps*		*Darlise*	
Compounds	February 2024	April 2024	February 2024	April 2024
PHENOLIC ACIDS				
*Cinnamic acid*				
Chlorogenic acid	37.80 ± 0.17	357.96 ± 0.40	145.12 ± 0.79	444.48 ± 0.34
Ferulic acid	16.10 ± 0.12	27.10 ± 1.47	19.12 ± 1.00	24.68 ± 0.40
Quinic acid	186.60 ± 2.21	171.56 ± 6.62	190.40 ± 2.26	232.20 ± 2.09
**SUM**	**240.50 ± 2.50**	**556.62 ± 8.49**	**354.64 ± 4.05**	**703.36 ± 2.83**
FLAVONOIDS				
*Flavones*				
Luteolin	*nf*	<LOQ	*nf*	<LOQ
Kaempferol	*nf*	<LOQ	*nf*	<LOQ
**SUM**	**-**	**-**	**-**	**-**
*Flavanons*				
Naringin	0.84 ± 0.06	0.56 ± 0.23	1.72 ± 0.06	0.76 ± 0.06
**SUM**	**0.84 ± 0.06**	**0.56 ± 0.23**	**1.72 ± 0.06**	**0.76 ± 0.06**
*Flavonols*				
Isoquercetin	127.07 ± 1.05	40.84 ± 0.06	113.56 ± 0.71	31.64 ± 0.62
Rutin	705.28 ± 2.10	1966.00 ± 2.60	347.32 ± 0.40	2042.44 ± 2.88
Isorhamnetin-3-rutinoside	100.60 ± 0.28	123.80 ± 1.19	32.32 ± 0.11	107.76 ± 1.24
**SUM**	**932.95 ± 3.43**	**2130.64 ± 3.85**	**493.20 ± 1.22**	**2181.84 ± 4.74**
**TOTAL POLYPHENOLS**	**1174.29 ± 5.99**	**2687.82 ± 12.57**	**849.96 ± 5.39**	**2885.96 ± 7.63**

**Table 4 foods-14-02367-t004:** Polyphenolic content (mg/kg dry weight) in aqueous extracts of cooked-digested portions of asparagus spears from two cultivars (*Placoseps* and *Darlise*) harvested in February and April 2024, analyzed by UHPLC Q-Orbitrap HRMS. Values are expressed as mean ± standard deviation.

	Cooked-Digested Portion (mg/kg) ± SD			
	*Placoseps*		*Darlise*	
Compounds	February 2024	April 2024	February 2024	April 2024
PHENOLIC ACIDS				
*Cinnamic acid*				
Chlorogenic acid	0.46 ± 0.07	13.50 ± 0.62	0.39 ± 0.03	20.65 ± 0.22
Ferulic acid	5.46 ± 0.00	28.91 ± 0.03	15.38 ± 0.00	26.29 ± 0.01
*p*-Coumaric acid	0.73 ± 0.02	7.71 ± 0.09	0.52 ± 0.08	5.93 ± 0.57
Quinic acid	68.94 ± 1.84	111.49 ±1.17	69.08 ± 0.22	121.91 ± 0.66
**SUM**	**75.59 ± 1.79**	**161.61 ± 1.92**	**85.37 ± 0.33**	**174.78 ± 1.47**
FLAVONOIDS				
*Flavanons*				
Naringin	0.14 ± 0.03	0.38 ± 0.04	*nf*	0.27 ± 0.04
**SUM**	**0.14 ± 0.03**	**0.38 ± 0.04**	* **nf** *	**0.27 ± 0.04**
*Flavonols*				
Isoquercetin	12.40 ± 0.00	0.92 ± 0.07	11.37 ± 0.00	1.37 ± 0.31
Rutin	43.60 ± 0.06	218.82 ± 0.06	45.21 ± 0.11	240.46 ± 3.03
Isorhamnetin-3-rutinoside	15.22 ± 0.10	38.55 ± 0.78	19.09 ± 0.00	37.71 ± 1.02
**SUM**	**71.22 ± 0.16**	**258.29 ± 0.91**	**75.67 ± 0.11**	**279.53 ± 4.36**
**TOTAL POLYPHENOLS**	**146.95 ± 1.98**	**420.286 ± 2.868**	**161.040 ± 0.441**	**454.581 ± 5.88**

**Table 5 foods-14-02367-t005:** Total phenolic content (TPC), measured using the Folin–Ciocalteu method, in the edible portion, cooked portion, and cooked-digested portion of *Asparagus officinalis* L. from two cultivars (*Placoseps* and *Darlise*), harvested in February and April 2024. Results are expressed as mg gallic acid equivalents (GAE) per kilogram of dry weight ± standard deviation (SD).

TPC-Edible Part (mg GAE/g) ± SD			
*Placoseps*		*Darlise*	
February 2024	April 2024	February 2024	April 2024
2.86 ± 0.04	7.01 ± 0.07	2.93 ± 0.01	5.89 ± 0.00
**TPC-Cooked portion (mg GAE/g) ± SD**			
* **Placoseps** *		* **Darlise** *	
**February 2024**	**April 2024**	**February 2024**	**April 2024**
6.34 ± 0.18	15.88 ± 0.16	6.03 ± 0.21	10.53 ± 0.00
**TPC-Cooked-digested portion (mg GAE/g) ± SD**			
* **Placoseps** *		* **Darlise** *	
**February 2024**	**April 2024**	**February 2024**	**April 2024**
1.65 ± 0.01	4.24 ± 0.08	1.70 ± 0.02	3.61 ± 0.01

**Table 6 foods-14-02367-t006:** Antioxidant capacity assessed by ABTS, DPPH, and FRAP assays in the edible portion, cooked portion, and cooked-digested portion of *Asparagus officinalis* L. from two cultivars (*Placoseps* and *Darlise*), harvested in February and April 2024. Results are expressed as mmol Trolox equivalents (TE) per kilogram of dry weight ± standard deviation (SD).

	Edible Part (mmol Trolox/kg) ± SD			
	*Placoseps*		*Darlise*	
	February 2024	April 2024	February 2024	April 2024
**DPPH**	6.74 ± 0.01	18.31 ± 0.06	10.48 ± 0.01	15.05 ± 0.06
**ABTS**	38.12 ± 0.15	64.14 ± 0.49	27.47 ± 0.27	32.99 ± 0.14
**FRAP**	13.26 ± 0.02	21.11 ± 0.08	19.15 ± 0.05	19.60 ± 0.09
	**Cooked portion (mmol Trolox/kg) ± SD**			
	* **Placoseps** *		* **Darlise** *	
	**February 2024**	**April 2024**	**February 2024**	**April 2024**
**DPPH**	10.23 ± 0.01	23.71 ± 0.22	12.13 ± 0.01	21.69 ± 0.03
**ABTS**	53.65 ± 0.04	67.69 ± 0.32	51.72 ± 0.36	59.88 ± 0.31
**FRAP**	15.71 ± 0.09	25.38 ± 0.11	17.83 ± 0.07	23.24 ± 0.03
	**Cooked–Digested portion (mmol Trolox/kg) ± SD**			
	* **Placoseps** *		* **Darlise** *	
	**February 2024**	**April 2024**	**February 2024**	**April 2024**
**DPPH**	1.56 ± 0.01	5.42 ± 0.01	2.44 ± 0.00	4.30 ± 0.02
**ABTS**	11.71 ± 0.13	18.12 ± 0.06	15.79 ± 0.68	14.22 ± 0.04
**FRAP**	10.05 ± 0.20	13.46 ± 0.05	12.46 ± 0.23	11.44 ± 0.03

## Data Availability

The original contributions presented in this study are included in the article/Appendix A. Further inquiries can be directed to the corresponding author.

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
