# Peer review of "Effect of Cooking and in vitro Digestion on the Polyphenols and Antioxidant Properties of Asparagus officinalis L. cultivars"

_foods, 2025, doi:10.3390/foods14132367_

Round 1
Reviewer 1 Report
Comments and Suggestions for Authors
This manuscript investigates the effects of cooking and in vitro digestion on the phytochemical composition and antioxidant properties of two asparagus varieties. The research design is comprehensive, the technical approach is clear, and the results are detailed. The study encompasses the complete chain from field to table to digestion, comparing different varieties and different harvest seasons, which increases the dimensionality and comparative value of the research. The study employs high-sensitivity UHPLC-Q-Orbitrap HRMS technology for polyphenol identification and quantification, with rigorous methodology. It also provides detailed chromatographic and mass spectrometric parameters for 27 polyphenolic compounds, as well as their content changes in edible portions, after cooking, and after digestion, demonstrating excellent data integrity. The innovation of this paper lies in its systematic examination of polyphenol component changes throughout the entire process from fresh asparagus to cooking to digestion, providing new evidence for understanding the mechanisms by which food processing and digestion affect bioactive substances. Overall, this research has significant academic value and application prospects. Specific revision suggestions are as follows:
1. The study only employed one cooking method; it is recommended to include other common household cooking methods such as steaming, baking, and microwaving for comparison to provide more comprehensive cooking guidance.
2. Some data fluctuations are significant but not fully explained, such as the rutin content of the Placoseps variety in Table 2.
3. In the introduction section, it is suggested to strengthen the discussion of the biological functions and health benefits of polyphenolic substances in asparagus, as well as existing research on nutritional differences in asparagus harvested in different seasons.
4. In the results section, add calculations of bioavailability percentages (ratio of polyphenol content after digestion to after cooking) and supplement the correlation analysis between polyphenol content and antioxidant activity.
5. In the conclusion section, include specific application recommendations, such as optimal cooking methods and best consumption seasons, as well as propose possible strategies to improve the bioaccessibility of asparagus polyphenols.
Author Response
Comments 1: This manuscript investigates the effects of cooking and in vitro digestion on the phytochemical composition and antioxidant properties of two asparagus varieties. The research design is comprehensive, the technical approach is clear, and the results are detailed. The study encompasses the complete chain from field to table to digestion, comparing different varieties and different harvest seasons, which increases the dimensionality and comparative value of the research. The study employs high-sensitivity UHPLC-Q-Orbitrap HRMS technology for polyphenol identification and quantification, with rigorous methodology. It also provides detailed chromatographic and mass spectrometric parameters for 27 polyphenolic compounds, as well as their content changes in edible portions, after cooking, and after digestion, demonstrating excellent data integrity. The innovation of this paper lies in its systematic examination of polyphenol component changes throughout the entire process from fresh asparagus to cooking to digestion, providing new evidence for understanding the mechanisms by which food processing and digestion affect bioactive substances. Overall, this research has significant academic value and application prospects. Specific revision suggestions are as follows:
The study only employed one cooking method; it is recommended to include other common household cooking methods such as steaming, baking, and microwaving for comparison to provide more comprehensive cooking guidance.
Response 1:
The authors thank the reviewer for this precious comment. We appreciate your suggestion to include multiple cooking methods for comparison. The aim of this work was to evaluate polyphenol bioaccessibility after digestion following the cooking process. In this study, boiling on an induction hob was selected as one of the most common household methods for cooking, and induction hob is increasingly widespread in domestic kitchens. However, this is an important suggestion for future research to further investigate the effect of other cooking methods, such as steaming, baking and microwaving, on bioaccessibility.
Comments 2: Some data fluctuations are significant but not fully explained, such as the rutin content of the Placoseps variety in Table 2.
Response 2: The authors thank the reviewer for this insightful observation. We acknowledge the significant fluctuations in certain data points, such as the rutin content of the Placoseps variety reported in Table 2. These variations may be attributed to biological variability among samples and environmental factors influencing the phytochemical composition, including seasonal changes that can enhance polyphenol biosynthesis and accumulation, which likely explain the marked increase in rutin content from February to April. The authors commented on the rutin content as follows: “The polyphenolic composition showed considerable variation among cultivars and harvest periods. Among the identified compounds, rutin consistently emerged as the most abundant in all analysed samples. In April, its concentration reached 1,770.72 mg/kg in Placoseps and 995.20 mg/kg in Darlise, representing around 70% of the total polyphenol content in both cultivars. Seasonal differences were evident across all major flavonols. In Placoseps, rutin levels increased by over 100-fold from February to April, while in Darlise, the increase was approximately six-fold”.
Comments 3: In the introduction section, it is suggested to strengthen the discussion of the biological functions and health benefits of polyphenolic substances in asparagus, as well as existing research on nutritional differences in asparagus harvested in different seasons.
Response 3: The authors thank the reviewer for this valuable comment. As suggested, the introduction has been revised to include a more detailed discussion on the biological functions and health benefits of polyphenolic compounds in asparagus, as well as existing research on the nutritional differences related to the harvest season.
Comments 4: In the results section, add calculations of bioavailability percentages (ratio of polyphenol content after digestion to after cooking) and supplement the correlation analysis between polyphenol content and antioxidant activity.
Response 4: The authors thank the reviewer for this constructive comment. As suggested by the reviewer, we have added a correlation analysis between total polyphenol content (TPC) and antioxidant activity (DPPH, ABTS, and FRAP) to provide a more comprehensive understanding of their interaction. These results are now included in the Supplementary Material (Table S1). Regarding bioavailability, the authors recognize its importance, and it will be addressed in future investigations.
Comments 5: In the conclusion section, include specific application recommendations, such as optimal cooking methods and best consumption seasons, as well as propose possible strategies to improve the bioaccessibility of asparagus polyphenols.
Response 5: The authors thank the reviewer for this helpful suggestion. In response, the Conclusion section has been revised to include specific application recommendations, such as the most suitable cooking method and the best season for asparagus consumption based on polyphenol content and antioxidant activity. Furthermore, we have proposed possible strategies to enhance the bioaccessibility of asparagus polyphenols, including the use of encapsulation techniques - both acid-resistant and conventional - as well as other optimized processing methods.
- Response to Comments on the Quality of English Language
Point 1: The English could be improved to more clearly express the research.
Response 1: The authors thank the reviewer for this observation. In response, the manuscript has been carefully revised to improve the clarity and quality of the English, ensuring that the research is communicated more effectively.

Reviewer 2 Report
Comments and Suggestions for Authors
The manuscript “Impact of Cooking and in vitro Digestion on the Phytochemical Profile and Antioxidant Properties of Asparagus officinalis L. cultivars presents a well-structured and comprehensive study assessing how cooking and in vitro digestion affect the polyphenolic content and antioxidant activity in two cultivars of Asparagus officinalis. The study uses robust analytical techniques (UHPLC-Q-Orbitrap HRMS) and standardized digestion protocols (INFOGEST), making the data highly reliable and relevant for food science and nutrition.
The manuscript is scientifically sound but requires revision to improve clarity, grammar, and logical flow. A full list of in-text comments and suggestions has been provided in the attached PDF to assist the authors with revision.
- The introduction is informative and places the study in context well. However, there is narrowing focus to emphasize the knowledge gap your study addresses—particularly the lack of data on bioaccessibility after digestion.
- The experimental section is comprehensive and follows good standards.
- The discussion is strong and well-referenced.
- The conclusion is sound but can be shortened for focus.
- Emphasize future research directions more clearly, especially the need for bioavailability (not just bioaccessibility) studies and strategies to enhance polyphenol stability.
- Ensure consistent use of terms: “polyphenolic compounds” vs. “polyphenols”; “edible part” vs. “edible portion”.
-
- Use µg/g or mg/kg consistently and always clarify if it's dry or fresh weight

Throughout the manuscript, passive structures and awkward phrasing should be revised.
Author Response
The authors confirm that all reviewer comments have been carefully addressed and clearly marked within the revised manuscript (PDF file).

Reviewer 3 Report
Comments and Suggestions for Authors
Overall, this manuscript presents an interesting and scientifically sound topic, supported by appropriate evidence and methodology. However, the narrative would benefit from improved coherence and a more engaging presentation of the study’s background and significance. Strengthening the storytelling flow and enhancing the logical connections between sections will help draw the reader in and highlight the relevance and impact of the research more effectively. The following comments are provided to support this improvement.
The term "phytochemical" in the title and keywords is too broad. Replacing it with "polyphenol" would improve specificity and enhance searchability, especially for readers focused on antioxidant research.
Including a schematic in the Introduction to show the relationship between the edible part, cooked portion, and cooked-digested portion would help readers better understand the rationale and importance of tracking bioactive changes through these stages.
Please revise the objective to better highlight the impact of the cooking process and in vitro digestion, so as to engage readers more effectively and clearly convey the core focus and significance of the study.
The authors are encouraged to clarify the clearly rationale behind choosing February and April as the sample collection periods. What specific environmental or agronomic factors during these months could influence the polyphenol content? Furthermore, please expand the Discussion to address how seasonal variation may affect the biosynthesis or accumulation of polyphenols, and explain the underlying mechanisms or physiological responses involved.
Please consider using the term "polyphenol" instead of "phenolic compound" in line 230 to maintain consistency and avoid potential confusion. Since the manuscript predominantly uses "polyphenol" elsewhere, uniform terminology would improve clarity and scientific precision
It appears that lines 234–239 focus specifically on the structural identification of isomeric compounds. However, it would improve clarity and scientific flow if the authors first provided a brief overview of the general identification approach used for all compounds (e.g., based on retention time, MS fragmentation patterns, or comparison with standards), and then elaborated specifically on how isomeric compounds were distinguished. This structure will help readers better understand the overall identification strategy and how isomers were treated as a special case.
The manuscript refers to "edible part" and "non-edible part" of the sample. For clarity and reproducibility, please specify which anatomical parts of the plant or sample these terms refer to.
The manuscript briefly mentions potential applications, but the Discussion would benefit from a clearer explanation of the importance of phenolic reduction after digestion, especially regarding nutritional and functional implications. Please elaborate on this point and support it with relevant literature. Expanding the literature review to include related studies would also help contextualize the findings and strengthen the study’s rationale.
Author Response
- Point-by-point response to Comments and Suggestions for Authors
Comment 1: Overall, this manuscript presents an interesting and scientifically sound topic, supported by appropriate evidence and methodology. However, the narrative would benefit from improved coherence and a more engaging presentation of the study’s background and significance. Strengthening the storytelling flow and enhancing the logical connections between sections will help draw the reader in and highlight the relevance and impact of the research more effectively. The following comments are provided to support this improvement.
The term "phytochemical" in the title and keywords is too broad. Replacing it with "polyphenol" would improve specificity and enhance searchability, especially for readers focused on antioxidant research.
Response 1: As suggested by the reviewer, we have replaced the term "phytochemical" with "polyphenol" in the title.
Comment 2: Including a schematic in the Introduction to show the relationship between the edible part, cooked portion, and cooked-digested portion would help readers better understand the rationale and importance of tracking bioactive changes through these stages.
Response 2: In response to the reviewer’s suggestion, the authors have prepared and included a graphical representation highlights the relationship between the edible part, the cooked portion, and the cooked-digested portion, helping to clarify the rationale of the study.
Comment 3: Please revise the objective to better highlight the impact of the cooking process and in vitro digestion, so as to engage readers more effectively and clearly convey the core focus and significance of the study.
Response 3: As suggested by the reviewer, the objective of the study has been revised to better emphasize the impact of the cooking process and in vitro digestion.
Comment 4: The authors are encouraged to clarify the clearly rationale behind choosing February and April as the sample collection periods. What specific environmental or agronomic factors during these months could influence the polyphenol content? Furthermore, please expand the Discussion to address how seasonal variation may affect the biosynthesis or accumulation of polyphenols, and explain the underlying mechanisms or physiological responses involved.
Response 4: We thank the reviewer for this valuable comment. In response, we have clarified the rationale behind choosing February and April as the sample collection periods, highlighting relevant environmental and agronomic factors that may influence polyphenol content. Furthermore, we have expanded the Discussion section to address how seasonal variation can affect the biosynthesis and accumulation of polyphenols.
Comment 5: Please consider using the term "polyphenol" instead of "phenolic compound" in line 230 to maintain consistency and avoid potential confusion. Since the manuscript predominantly uses "polyphenol" elsewhere, uniform terminology would improve clarity and scientific precision
Response 4: We appreciate this insightful suggestion. To maintain consistency and improve clarity, we have replaced the term “phenolic compound” with “polyphenol” in line 230, in alignment with the terminology used throughout the manuscript.
Comment 6: It appears that lines 234–239 focus specifically on the structural identification of isomeric compounds. However, it would improve clarity and scientific flow if the authors first provided a brief overview of the general identification approach used for all compounds (e.g., based on retention time, MS fragmentation patterns, or comparison with standards), and then elaborated specifically on how isomeric compounds were distinguished. This structure will help readers better understand the overall identification strategy and how isomers were treated as a special case.
Response 6: We thank the reviewer for this thoughtful and constructive comment. Accordingly, we have revised lines 234–239 to first describe the general approach used for all compounds—based on retention time, MS fragmentation patterns, and comparison with authentic standards.
Comment 7: The manuscript refers to "edible part" and "non-edible part" of the sample. For clarity and reproducibility, please specify which anatomical parts of the plant or sample these terms refer to.
The manuscript briefly mentions potential applications, but the Discussion would benefit from a clearer explanation of the importance of phenolic reduction after digestion, especially regarding nutritional and functional implications. Please elaborate on this point and support it with relevant literature. Expanding the literature review to include related studies would also help contextualize the findings and strengthen the study’s rationale.
Response 7: We thank the reviewer for this valuable and insightful comment. In response, we have clarified the terms "edible part" and "non-edible part" by specifying the corresponding anatomical parts of the plant to improve clarity and reproducibility. Furthermore, we have expanded the Discussion section to provide a more detailed explanation of the nutritional and functional implications of the observed reduction in polyphenol content after digestion.
Round 2
Reviewer 3 Report
Comments and Suggestions for Authors
The revised manuscript reflects meaningful improvements in response to the reviewer comments. The authors have effectively clarified key points, strengthened the narrative flow, and addressed the study’s significance more clearly. I consider the current version acceptable for publication.